# Amide-Containing Bottlebrushes via Continuous-Flow Photoiniferter Reversible Addition–Fragmentation Chain Transfer Polymerization: Micellization Behavior

**DOI:** 10.3390/polym16010134

**Published:** 2023-12-31

**Authors:** Alexey Sivokhin, Dmitry Orekhov, Oleg Kazantsev, Ksenia Otopkova, Olga Sivokhina, Ilya Chuzhaykin, Alexey Ovchinnikov, Olga Zamyshlyayeva, Irina Pavlova, Olga Ozhogina, Maria Chubenko

**Affiliations:** 1Research Laboratory “New Polymeric Materials”, Nizhny Novgorod State Technical University, n.a. R.E. Alekseev, 24 Minin Street, 603155 Nizhny Novgorod, Russia; 2V.A. Kargin Research Institute of Chemistry and Technology of Polymers with Pilot Plant, 606000 Dzerzhinsk, Nizhegorodskaya obl., Russia; 3Department of High Molecular Compounds and Colloidal Chemistry, Faculty of Chemistry, Lobachevsky State University, Gagarina pr. 23, 603950 Nizhny Novgorod, Russia

**Keywords:** continuous flow, photoiniferter polymerization, thermoresponsive bottlebrushes, self-assembly, self-folding, hydrogen bonding

## Abstract

Herein, a series of ternary amphiphilic amide-containing bottlebrushes were synthesized by photoiniferter (PI-RAFT) polymerization of macromonomers in continuous-flow mode using trithiocarbonate as a chain transfer agent. Visible light-mediated polymerization of macromonomers under mild conditions enabled the preparation of thermoresponsive copolymers with low dispersity and high yields in a very short time, which is not typical for the classical reversible addition–fragmentation chain transfer process. Methoxy oligo(ethylene glycol) methacrylate and alkoxy(C_12_–C_14_) oligo(ethylene glycol) methacrylate were used as the basic monomers providing amphiphilic and thermoresponsive properties. The study investigated how modifying comonomers, acrylamide (AAm), methacrylamide (MAAm), and N-methylacrylamide (-MeAAm) affect the features of bottlebrush micelle formation, their critical micelle concentration, and loading capacity for pyrene, a hydrophobic drug model. The results showed that the process is scalable and can produce tens of grams of pure copolymer per day. The unmodified copolymer formed unimolecular micelles at temperatures below the LCST in aqueous solutions, as revealed by DLS and SLS data. The incorporation of AAm, MAAm, and N-MeAAm units resulted in an increase in micelle aggregation numbers. The resulting bottlebrushes formed uni- or bimolecular micelles at extremely low concentrations. These micelles possess a high capacity for loading pyrene, making them a promising choice for targeted drug delivery.

## 1. Introduction

In recent decades, polymer science has concentrated on achieving the accurate synthesis of well-defined macromolecules with a specific architecture and molecular weight characteristics. The discovery of reversible-deactivation radical polymerization techniques marked a significant advancement in this field. Visible light-mediated photo-induced reversible addition–fragmentation chain transfer (RAFT) polymerization, which is gaining popularity due to a number of benefits over the classical RAFT process, is a further development of this approach. These advantages include [1,2,3,4,5,6,7,8,9,10,11] the use of low-cost LED light sources, polymerization under ambient conditions, and the use of eco-friendly solvents, e.g., water. Achieving spatiotemporal control and high chain-end fidelity enables easy and efficient production of block copolymers. Photo-induced RAFT polymerization can occur through three different methods. The first method is conventional RAFT polymerization, which requires a chain transfer agent and photoinitiator to act as a radical source. The second method is photoiniferter polymerization (PI-RAFT), where the RAFT agent itself acts as the initiator, decomposing into free radicals under specific wavelengths of light. The third method is PET-RAFT polymerization, which does not require an external radical initiator because the photoredox catalyst’s excited state can directly activate the thiocarbonylthio compounds through photoinduced electron or energy transfer, initiating polymerization [2,12,13,14,15,16,17,18].

Molecular oxygen is known to inhibit radical polymerization. Photoinduced polymerization is extremely sensitive to the presence of oxygen. Together with the strong attenuation of light within the reaction medium due to absorption, this can lead to a complete stoppage of the polymerization, as well as the impossibility of scaling the process. In some cases, common methods of oxygen removal (freeze–pump–thaw cycles, nitrogen purging) may yield unsatisfactory outcomes.

The oxygen-tolerant versions of PET-RAFT polymerization have significantly improved the situation. In addition, enzyme-assisted polymerization variants, for example in the presence of glucose oxidase and glucose, have gained popularity in recent years [19,20,21,22,23,24,25,26]. Glucose oxidase can scavenge oxygen and has been used in RDRP processes for its removal. The enzyme oxidizes glucose to δ-gluconolactone (which then spontaneously hydrolyzes to gluconic acid) in the presence of molecular oxygen, producing hydrogen peroxide. However, the use of expensive and sometimes toxic photocatalysts or enzymes requires further careful purification of the polymers if they are to be used for medical purposes.

Photoiniferter polymerization, unlike PET-RAFT, is not oxygen-tolerant; however, the use of a continuous-flow photoreactor together with a suitable solvent makes it possible to achieve very high conversions in a short time (2–3 h) while maintaining satisfactory control. In this case, the scalability of the process is also an additional important advantage.

Several studies have demonstrated the benefits of utilizing continuous-flow reactors over the traditional periodic mode in photomediated atom transfer radical polymerization processes. The limitations of the periodic mode include light attenuation in the reaction mixture and inability to scale the process. On the contrary, the scalable process [27,28,29], improved heat and mass transfer properties [28,30], increased light irradiation efficiency [31], higher conversions in less time [13,17,29,32,33,34,35,36], narrow molecular weight distribution [17,32], and simplified multi-step syntheses are some of the advantages of continuous-flow reactors [14,28,36,37,38,39,40]. These reactors allow for the sequential or parallel connection of multiple reactors, which simplifies complex syntheses. Overall, the utilization of continuous-flow reactors in photoRAFT processes has demonstrated encouraging outcomes and presents a plausible solution to the limitations of the periodic mode.

Thermoresponsive polymers with a low critical solution temperature (LCST) in aqueous solutions have gained attention for their potential use as smart materials in various applications [41,42,43,44,45]. Methoxy oligo(ethylene glycol) methacrylate (MOEGM) copolymers are promising candidates in this field because of their desirable properties such as biocompatibility, low toxicity, and biodegradability [46,47]. The LCST of these copolymers may be precisely tuned by modifying the quantity of ethoxylated moieties and selecting suitable hydrophobic comonomers. They demonstrate a sharp and reversible LCST transition with minimal hysteresis [48,49,50,51,52,53,54]. The recent literature [55,56,57,58] has emphasized the supramolecular assembly, thermoresponsive property control, and conformational management of PEG-based brushes. Studies have tested different derivatives of hydrophobically modified MOEGM polymers, revealing their capacity to form micellar structures in aqueous solutions. These polymers show promise as polymeric nanocontainers for delivering hydrophobic drugs. Furthermore, PEG-based brushes have an advantage in that they create unimolecular micelles [59,60,61,62], which offer greater stability in changing environmental conditions. Overall, these recent studies shed light on the fascinating characteristics and promising uses of hydrophobically modified MOEGM polymers.

Here, we report on the synthesis of ternary amphiphilic copolymers. The base that provided these copolymers with thermoresponsive properties consisted of methoxy oligo(ethylene glycol) methacrylate and alkoxy(C12–C14) oligo(ethylene glycol) methacrylate units. Additionally, units of modifying amide comonomers including acrylamide, methacrylamide, and N-methylacrylamide were incorporated. The aim of this study was to investigate the synthesis and self-assembly of these copolymers into micelle-like structures and to reveal the role of amide comonomers in these processes.

## 2. Materials and Methods

### 2.1. Materials

The starting monomers were acrylamide, methacrylamide, N-methylacrylamide, and methoxy oligo(ethylene glycol) methacrylate (MOEGM, Mn = 500) from Sigma-Aldrich (Moscow, Russia) and alkoxy(C12–C14) oligo(ethylene glycol) methacrylate (AOEGM) (Figure 1) synthesized following the procedure outlined in [63]. MOEGM and AOEGM were subsequently purified from an inhibitor by passing through basic alumina. The chain transfer agent CDTPA (4-cyano-4-[(dodecylsulfanylthiocarbonyl)sulfanyl]pentanoic acid) was synthesized following the method described in ref. [64]. All solvents, including dimethyl sulfoxide, tetrahydrofuran (ACS reagent, ≥99.5%), and acetonitrile (for spectroscopy, ≥99.5%) from Aldrich (Moscow, Russia) were utilized without purification.

### 2.2. Photoiniferter RAFT Polymerization

A continuous-flow reactor was used for the PI-RAFT polymerization. The reactor was composed of an aluminum cylinder with a diameter of 12 cm and a height of 8 cm that contained a LED strip. Inside the aluminum cylinder, a glass cylinder with a diameter of 7.5 cm was positioned coaxially with a PTFE tube of internal/external diameter of 2/3 mm wrapped around it. The LED strip was positioned 1.6 cm away from the surface of the tube. With an operating volume of 18.5 cubic centimeters, the irradiated part of the tubular reactor was 5.9 m long. The 5050 SMD LEDs (Wenzhou Rockgrand Trade Co., Ltd., Wenzhou, China) were utilized as light sources, with 60 LEDs per meter and a maximum power output of 14.4 W/m at 12 V. These LEDs emitted blue light with a wavelength of 470 nm. The light intensity was regulated by a switching power supply model PS3005N manufactured by QJE (Xinyujie Electronics Co., LTD., Shenzhen, China). Its value was assessed using an OHSP-350C spectral analyzer from Hangzhou Hopoo Light&Color Technology Co., LTD, Hangzhou, China, and adjusted to 5 mW/cm^2^ nearby the surface of the tube.

PI-RAFT polymerization was performed as described below. CDTPA (14.7 mg, 35.3 μmol, 1.0 eq), MOEGM (1.2993 g, 2.81 mmol, 80 eq), AOEGM (1.7191 g, 2.81 mmol, 80 eq), and AAm (0.1008 g, 1.42 mmol, 40 eq) were mixed with DMSO or THF (4.23 g) and stirred until fully dissolved. The total monomer concentration in resulting mixture was 50 wt%. The experimental setup was assembled according to Figure 2. The reaction mixture in the feed vial was purged with nitrogen for 10 min. The tubular reactor and the collecting vial were then purged for five minutes. A 50 mL syringe prefilled with nitrogen was placed in the syringe pump and the reaction mixture was transferred to the tubular reactor by pumping nitrogen at the desired rate. To protect the collecting vial from light, it was covered with foil. The syringe pump regulated the residence time of the reagents in the reactor. A product aliquot was mixed with acetonitrile to analyze the monomer conversion using HPLC. The polymerization was stopped by exposing the mixture to air and cooling it down in the dark. The copolymers were further diluted with tenfold ethyl alcohol and purified by dialysis (MWCO 8–14k) in ethyl alcohol for three days in the dark and then dried under vacuum. Their structures (Figure 3) were confirmed by ^1^H NMR (Appendix A) and IR spectroscopy. ^1^H NMR [400 MHz, chloroform-d, 25 °C, δ = 7.27 (chloroform)]: δ, 4.06 (COOCH_2_-), 3.71–3.42 (-CH_2_O(CH_2_CH_2_O)_n_CH_2_-, -CH_2_O(CH_2_CH_2_O)_n_CH_3_), 3.35 (-CH_2_O(CH_2_CH_2_O)_n_CH_3_), 2.1–1.65 (-CH_2_C(CH_3_)-), 1.55 (-OCH_2_CH_2_(CH_2_)_m_CH_3_), 1.24 (-OCH_2_CH_2_(CH_2_)_m_CH3), 1.0–0.86 ((-CH_2_C(CH_3_)-, -OCH_2_CH_2_(CH_2_)_m_CH_3_).

### 2.3. Characterization Techniques

^1^H NMR spectra were recorded at 25 °C in CDCl_3_ or DMSO-d6 using an Agilent 400 MHz DD2 spectrometer (Agilent Technologies, Santa Clara, CA, USA). The dn/dc values for copolymers were determined within a concentration range of 1–15 mg/mL at 27–30 °C using a BI-DNDC differential refractometer from Brookhaven Instr. Corp., Holtsville, NY, USA. The monomer concentrations in reaction mixtures were measured using an HPLC system that included a Kromasil 100-5-C18 4.6 × 250 mm column, refractometric and matrix UV detectors, a thermostat manufactured by Shimadzu Prominence (Tokyo, Japan). Acetonitrile was used as an eluent, with a flow rate of 0.9 mL/min. The thermostat temperature was of 55 °C.

Polymer molecular weights and molecular weight distributions were determined through GPC analysis, utilizing a Chromos LC-301 instrument (Chromos, Dzerzhinsk, Russia) equipped with an Alpha-10 isocratic pump and a Waters 410 refractometric detector, along with two exclusive columns, Phenogel 5 µm 500A and Phenogel 5 µm 10E5A, from Phenomenex (with a measuring range of 1 k to 1000 k); tetrahydrofuran was used as the eluent. Calibration was performed using polystyrene standards.

Differential scanning calorimetry (DSC) was conducted for polymer specimens (approximately 10–15 mg in an aluminum pan) under a dry argon flow utilizing a DSC 204F1 Phoenix calorimeter (Netzsch, Selb, Germany) furnished with a CC 200 controller for liquid nitrogen cooling. The heating and cooling rates were set at 10 °C/min and −10 °C/min, respectively, between −80 °C and 80 °C.

Laser light scattering (LLS) experiments were conducted using a Photocor Complex multi-angle light-scattering device (Photocor Ltd., Moscow, Russia) that was equipped with a thermostabilized diode laser (λ = 659 nm, 35 mW) and a thermo-electric Peltier temperature controller (temperature range from 5 to 100 °C, accuracy of 0.1 °C). LLS was employed to measure the hydrodynamic radii (Rh) of polymer molecules and micelles (DLS), weight-averaged molecular weights (Mw), second virial coefficients (A2), and aggregation numbers (N_agg_) of micelles (SLS).

After preparation, the polymer solutions were equilibrated at room temperature for 24 h and filtered using CHROMAFIL PET syringe filters (0.20 μm) before conducting the measurements. At least three measurements were conducted per sample, resulting in an average hydrodynamic radius of Rh in nanometers. The single-angle Debye plot method was utilized to determine Mw and A2.

The scattering geometry employed a vertically polarized incident light and detection without a polarizer (VU geometry, Rv). According to [65], the Rayleigh ratio for toluene at an incident wavelength of 659 nm and measurement temperature was calculated to be 1.142 × 10^−5^ cm^−1^ at 25 °C.

Critical micelle concentrations (CMCs) of copolymers were determined using pyrene as a fluorescent probe via fluorimetry. To obtain the copolymer solutions, 10 different concentrations ranging from 1 × 10^−6^ to 0.5 mg/mL were prepared by dissolving the polymers in an aqueous pyrene solution (6 × 10^−7^ M). The resulting mixtures were sonicated for 5 min and then incubated at room temperature for 24 h prior to the measurements. Steady-state fluorescence spectra were measured on a Shimadzu RF-6000 spectrofluorometer (Shimadzu, Tokyo, Japan) under specified conditions: excitation slit width of 3 nm, emission slit width of 3 nm, scanning speed of 200 nm/min, excitation wavelength of 335.0 nm, and emission wavelength of 350.0–500.0 nm. The ratio of intensities of the first (I_1_, 373 nm) and third (I_3_, 384 nm) vibronic pyrene emission bands, denoted as (I_1_/I_3_), was measured as a function of copolymer concentration. The concentration corresponding to the inflection point at which I_1_/I_3_ begins to decrease was defined as CMC.

Additionally, pyrene, a hydrophobic drug model, was employed to assess the loading capacity of the micelles. The loading capacity was evaluated through UV spectroscopy using the subsequent procedure. An amount of 5 milligrams of dry pyrene was added to 10 milliliters of a 0.1% aqueous polymer solution, and the resulting solution was sonicated at 25 °C for 30 min. The aqueous polymer solution containing pyrene was then filtered using a syringe filter with a pore size of 0.45 μm. The filtrate was diluted 40-fold with acetonitrile. The concentration of pyrene was measured using a Shimadzu UV-1800 (Shimadzu, Tokyo, Japan) spectrophotometer at an absorption wavelength of 334 nm, with a standard calibration curve experimentally determined for pyrene solutions in acetonitrile.

## 3. Results and Discussion

### 3.1. Photoiniferter RAFT Copolymerization

The objective of this study was to develop a simple and efficient method for creating thermoresponsive nonionic copolymers that can form micelles in aqueous solutions and effectively release hydrophobic drugs at a controlled rate. MOEGM homopolymers with side chains of various lengths are known to exhibit thermoresponsive properties [50,51,66]. We used MOEGM and hydrophobic comonomer with higher alkyl moieties (AOEGM) as the main polymer backbone to enhance the amphiphilic properties. Additionally, we introduced modifying comonomer units (acrylamide, methacrylamide and N-methylacrylamide) to assess their impact on self-assembly behavior, CMC, and drug loading capacity of micelles. The copolymers were prepared using the grafting through method. Previous research had determined the optimal conditions for obtaining similar copolymers, including the composition, molecular weight, and compositional homogeneity [62].

Through light-scattering techniques, it was determined that the copolymers could form unimolecular micelles in aqueous solutions, but only above a certain molecular weight where chain flexibility was sufficient. This was crucial for maintaining micelle stability upon dilution. Below this threshold, the copolymers formed multimolecular micelles even before the lower critical solution temperature (LCST). At temperatures above the LCST, the assemblies transformed into larger aggregates.

A methodology was developed for producing self-assembled copolymers with high chain-end fidelity by PI-RAFT polymerization in periodic mode. However, we encountered difficulties in scaling up the synthesis process. When using reaction vessels larger than 5–10 mL, polymerization either did not occur or had a significant induction period. This resulted in low product yields. To address this issue, we employed a flow-type reactor, which allowed obtaining high-molecular-weight copolymers with low dispersity and high conversions in a relatively short time.

The copolymers were synthesized using a reversible addition–fragmentation chain transfer (RAFT) agent, specifically 4-cyano-4-[(dodecylsulfanylthiocarbonyl)sulfanyl]pentanoic acid (CDTPA). Its structure, along with the mechanism of visible light-induced PI-RAFT polymerization, is depicted in Figure 4.

RAFT agents can be activated by UV or visible light. The choice of wavelength for irradiation depends on the type of chain transfer agent (CTA) used. The PI-RAFT polymerization is initiated when the C-S bond undergoes homolytic dissociation, leading to the cleavage of the leaving group. Photoexcitation can occur through either 𝜋-𝜋* or n-𝜋* transitions. Although the n-𝜋* transition is weaker, it is preferred for initiation because it causes fewer side reactions. The wavelength associated with the n-𝜋* transition differs depending on the type of CTA, falling in the UV region for xanthates and dithiocarbamates and in the visible region for trithiocarbonates and dithiobenzoates [67]. CDTPA used in this study can also be activated by UV light through the 𝜋-𝜋* transition (Figure 5), and green light is also acceptable. However, blue light provides an optimum balance between rate and control, making it the preferred choice.

Table 1 shows the polymerization results. The primary objective was to achieve the highest possible monomer conversion while maintaining acceptable dispersity, with Đ <1.3. The fastest reaction rate was found with DMSO solvent, which resulted in 91% conversion. As AAm and MAAm are less soluble in DMSO, the ternary copolymers were obtained in THF. In general, DMSO stands out as the most effective solvent for photo-mediated processes. This may be due to several reasons. It is believed that in PET-RAFT processes, fast deactivation of the reactive ground-state oxygen to the singlet species may be due to triplet–triplet annihilation with the excited-state photocatalyst, followed by a reaction with DMSO to form dimethylsulfone [4,8]. However, the reasons for the efficiency of DMSO in PI-RAFT processes are not completely clear.

The discrepancy between the theoretical molecular masses and the GPC data is noteworthy, with the latter being significantly underestimated. This has been observed multiple times for bottlebrushes based on macromonomers containing oligo(ethylene glycol) moieties [59,68,69,70,71], except for low-molecular-weight brushes [7,14,72]. The primary cause of this discrepancy might be the nonlinear relationship between retention time and molecular weight in the GPC resulting from changes in the shape and hydrodynamic volume of macromolecules with an increasing degree of polymerization (transition from a spherical to a worm-like structure). At the same time, the differences in hydrodynamics increase significantly compared to polystyrene standards.

The process is substantially slowed down by adding amide comonomers, particularly with the use of acrylamide instead of methacrylamide. The involvement of amide units in the copolymer is confirmed by IR spectroscopy data (Figure 6). The spectrum of the MOEGM-AOEGM copolymer contains characteristic bands corresponding to the stretching vibration of the carbonyl group (C=O, 1729 cm^−1^), asymmetric stretching vibration of ether bonds (C–O–C, 1115 cm^−1^). The peaks at 2861 cm^−1^ and 2925 cm^−1^ are the stretching vibrational bands of methylene (–CH_2_) and methyl (–CH_3_) groups. A broad band at 3300–3700 cm^−1^ with a maximum at 3518 cm^−1^ is attributed to the OH vibrations of hydrogen-bonded water; the band at 1642 cm^-1^ is the bend (v2) of liquid absorbed water. The spectra of ternary amide-containing copolymers additionally contain bands at ~3360 and 3208 cm^−1^ attributed to NH and NH_2_ stretching vibrations. The absorption bands at 1678, 1536 and 1293 cm^−1^ are attributed to C=O (amide) stretching, NH bending and C-N stretching vibrations, respectively.

The copolymers underwent thermal analysis through differential scanning calorimetry (DSC), displaying two types of transitions in the DSC thermograms, namely glass transition and melting. All numerical values are presented in Table 2. The glass transition temperature of the base copolymer was around −69 °C. The addition of up to 20% amide units resulted in an increase in Tg by 7–9 °C. All copolymers exhibited broadened peaks corresponding to the melting point (Tm) due to the presence of MOEGM units that are hard to crystallize (Figure 7). It is noteworthy that copolymers with acrylamide units displayed a distinct trend: the melting peak broadened and then split upon the introduction of 10% and 20% AAm shifting simultaneously to the higher-temperature region. Conversely, the melting peak became narrower when methacrylamide units were introduced.

### 3.2. Thermoresponsive Properties, Hydrodynamic and Molecular Weight Characteristics of Bottlebrushes

It is known that MOEGM-AOEGM copolymers exhibit thermoresponsive properties with the possibility of fine-tuning the LCST in a wide range by varying the ratio of hydrophilic (MOEGM) and hydrophobic (AOEGM) units. It was interesting to evaluate how the introduction of hydrophilic amide units would affect the LCST. C_p_ values were determined using turbidimetry (Table 3). It was surprising to find that in most cases, the addition of up to 20% amide units had little or no effect on thermoresponsive properties, shifting C_p_ by 1–3 °C in one direction or another.

The critical micelle concentrations (CMCs) of the copolymers were determined using pyrene as a fluorescent probe. Also, pyrene was used as a model of a hydrophobic drug when evaluating the drug loading capacity of the micelles.

In general, all copolymers demonstrate low critical micelle concentration due to their amphiphilic nature and high loading capacity values, regardless of the structure or content of the introduced amide groups. This can be attributed to the lack of centers capable of hydrogen bonding with amide groups in pyrene molecule, and the retention of pyrene in the micelles is mainly due to hydrophobic interactions.

Adjusting the hydrophilic–hydrophobic balance through modifications in the monomer structure and composition allows tuning the amphiphilic properties of the resulting molecular brushes and achieving such amphiphilic properties that ensure the formation of unimolecular micelles in aqueous solutions. The hydrophobic core of the unimolecular micelles consists of the hydrophobic main chain and hydrophobic side chains, while the hydrophilic side chains (or hydrophilic blocks of side chains) constitute the outer shell. The flexibility and capacity of macromolecules to form monomolecular micellar structures of the core–shell type depends on the length and composition of the hydrophobic main chain and hydrophilic or amphiphilic side chains. Molecular brushes can form unimolecular micelles in water through self-folding once they reach a certain degree of polymerization. Copolymers of similar composition with low molecular weight, that have limited flexibility, are compelled to form multimolecular micelles in order to reduce the contact surface of hydrophobic units with water since they are unable to fold.

Compositional homogeneity in polymers and narrow molecular weight distribution are crucial factors for the formation of unimolecular micelles, which in turn promote the unimodality of micelles. The compositional homogeneity was achieved by utilizing the similarly reactive oxyalkylated methacrylates with hydrophilic (MOEGM) and hydrophobic (AOEGM) moieties providing an amphiphilic nature. The RDRP polymerization methodology was employed to ensure optimal MW and dispersity.

In the present work, we tried to evaluate the effect of introducing a small amount of hydrophilic amide units capable of hydrogen bonding on the micelle aggregation number and the possibility of obtaining unimolecular micelles.

The micelle aggregation number in aqueous solutions was determined by analyzing copolymer solutions using static light scattering and dynamic light scattering. To calculate the number of macromolecules in a micelle in aqueous solution, Mw in water was divided by Mw in acetonitrile, assuming the latter as true.
N_agg_ = M_W_ (SLS in water)/M_W_ (SLS in a good or θ-solvent)(1)

Table 4 summarizes the data on the molecular weight characteristics of the bottlebrushes. The base copolymer P1, which contained no amide units, formed unimolecular micelles in water by self-folding with an aggregation number of about unity. The DLS data also confirm the existence of narrowly dispersed particles comparable in size to individual macromolecules in aqueous solutions. Comparing Rh values in water and acetonitrile indicates that all copolymers formed fairly dense micelles in water.

All amide-containing copolymers formed micelles with an aggregation number approaching two, with the highest values observed for the acrylic amide (AAm). This is likely a result of the ability to form a greater number of hydrogen bonds in comparison to the N-substituted amide, as well as significant differences in the reactivity of the monomers (when comparing AAm and MAAm), resulting in greater compositional heterogeneity in copolymers P4 and P5 compared to P2 and P3.

## 4. Conclusions

Amphiphilic thermoresponsive bottlebrushes composed of oligo(ethylene glycol)-containing macromonomers and up to 20% modifying amide comonomers were synthesized through photoiniferter polymerization in a continuous-flow photoreactor. The reaction, carried out at 40 °C under blue light for 1–2 h in DMSO and THF, produced high-molecular-weight copolymers with satisfactory dispersity (*Đ* < 1.3) and high yields. Considering the maximum achieved yield (91% for P1), this study demonstrates the potential to scale photopolymerization using the proposed setup to produce approximately 50 g of pure polymer over an 8 h day. The copolymers were characterized through various methods, including DSC, DLS, SLS, and turbidimetry. Thermal analysis indicated that the copolymers possessed a glass transition temperature ranging from approximately −60 to −70 °C, with an increase of 7–9 °C upon the introduction of 20% amide units compared to the base copolymer. The melting temperature increased only with the introduction of AAm units while remaining almost constant in the other cases. The unmodified copolymer formed unimolecular micelles at temperatures below the LCST in aqueous solutions, as shown by the DLS and SLS data. The addition of AAm, MAAm, and N-MeAAm units led to an increase in micelle aggregation numbers, with values ranging from 2 to 3 and AAm exhibiting the highest values. This is probably due to the increased susceptibility of AAm to hydrogen bonding and the compositional heterogeneity of such copolymers.

The resulting bottlebrushes possess easily modulated LCST and are capable of forming uni- or bimolecular micelles at very low concentrations. These micelles have sufficiently high loading capacity with respect to pyrene (a model of a hydrophobic drug) that hold potential for targeted drug delivery.

## Figures and Tables

**Figure 1 polymers-16-00134-f001:**
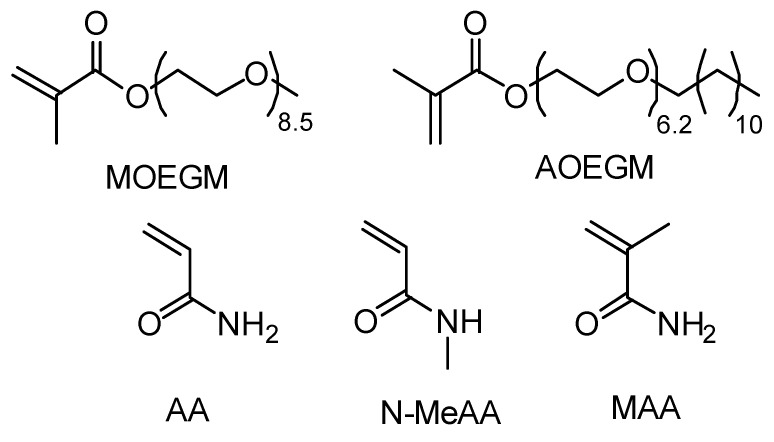
Monomer structures.

**Figure 2 polymers-16-00134-f002:**
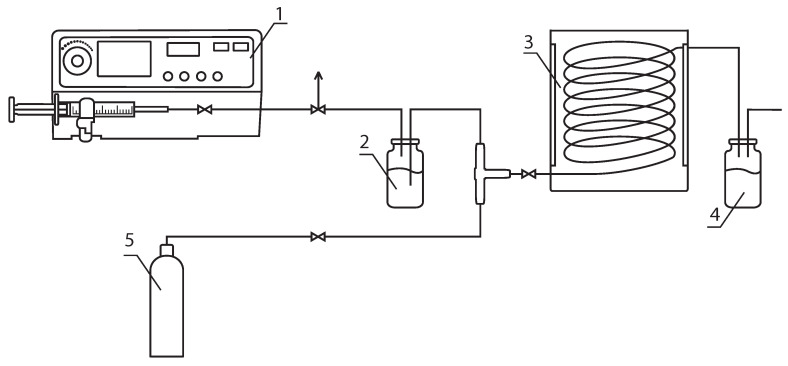
Experimental set-up for the PI-RAFT polymerization. 1—syringe pump; 2—feed vial; 3—photoreactor; 4—product vial; 5—nitrogen tank.

**Figure 3 polymers-16-00134-f003:**
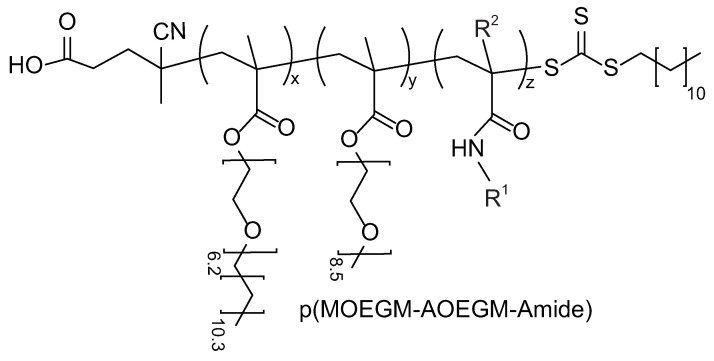
Structures of ternary p(MOEGM-AOEGM-Amide) copolymers. Amide units: acrylamide (R^1^ = H, R^2^ = H); methacrylamide (R^1^ = H, R^2^= CH_3_); N-methylacrylamide (R^1^ = CH_3_, R^2^ = H).

**Figure 4 polymers-16-00134-f004:**
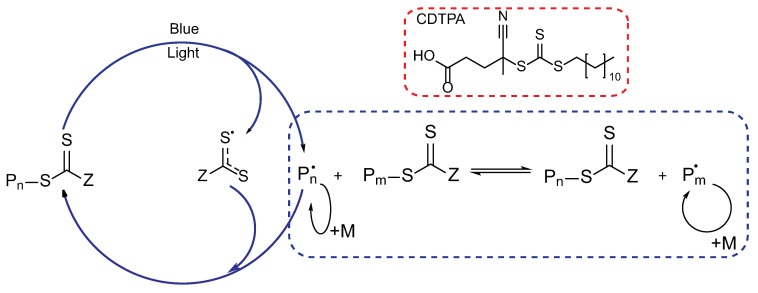
Structure of the RAFT agent and the mechanism of PI-RAFT polymerization under visible light.

**Figure 5 polymers-16-00134-f005:**
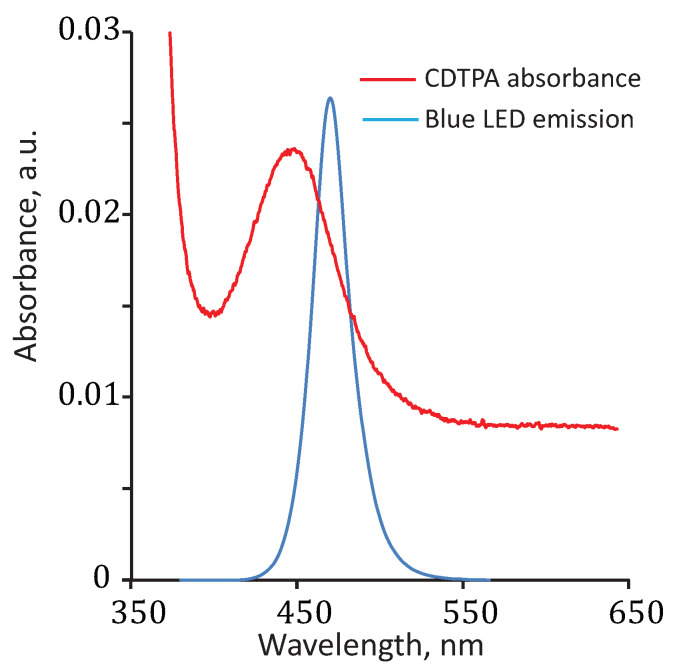
Absorption spectra of CDTPA measured using UV–Vis spectroscopy in acetonitrile (0.2 wt%), and emission spectra of blue LEDs. The peak of the emission spectra of blue LEDs is at 470 nm.

**Figure 6 polymers-16-00134-f006:**
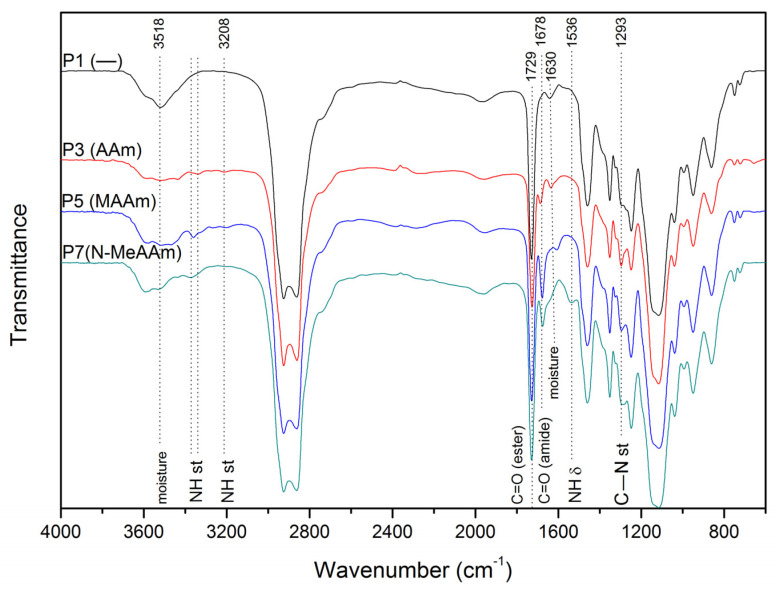
IR spectra of MOEGM-AOEGM-amide ternary copolymers. Sample designations are in Table 1.

**Figure 7 polymers-16-00134-f007:**
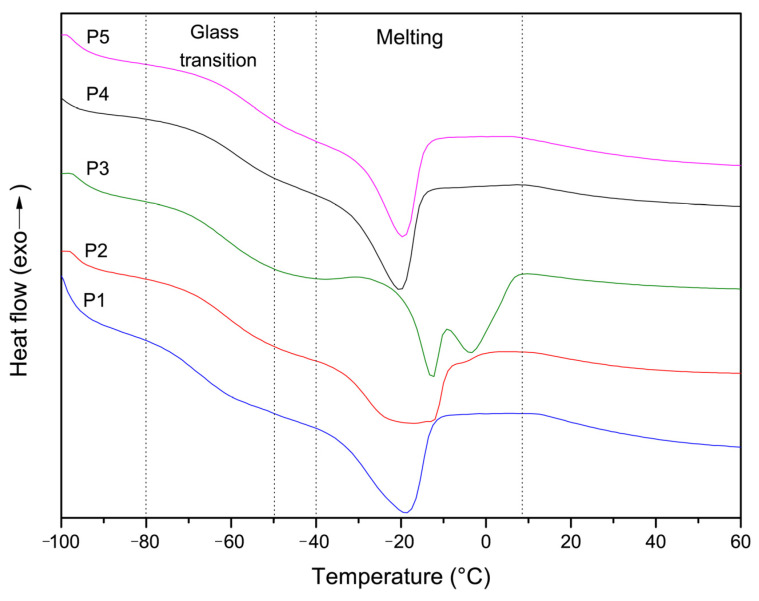
DSC thermograms for p(MPEGMA-AOEGM-amide)s. Heating rate: 10 °C/min.

**Table 1 polymers-16-00134-t001:** Characterization of (co)polymers synthesized via continuous-flow RAFT polymerization (concentration of monomers—50%, t = 40 °C, blue light intensity—5 mW/cm^2^, Σ[Monomers]_0_/[CDTPA]_0_ = 200:1).

ID	[MOEGM]_0_ /[AOEGM]_0_ /[M]_0_	Comonomer (M)	Solvent	Flow Rate, mL/min	Time, min	Conversion, %	Composition ^a^, m_1_:m_2_:m_3_ (mol)	M_n, th_ ^b^	M_n_ ^c^	M_w_ ^c^	Đ ^c^ (M_w_/M_n_)
P1	50:50:0	-	DMSO	9	137	91	49:51:0	95,400	8900	11,400	1.28
P2	45:45:10	AAm	THF	9	115	54	47.5:46.3:6.2	53,800	8500	10,500	1.23
P3	40:40:20	AAm	THF	9	111	42	43.2:44.3:12.4	39,800	8200	9900	1.21
P4	45:45:10	MAAm	THF	9	102	62	47.6:45.2:7.2	61,200	8100	9900	1.22
P5	40:40:20	MAAm	THF	9	111	50	44.5:41.6:13.8	46,600	6600	7800	1.18
P6	45:45:10	N-MeAAm	THF	8	124	55	48.3:46.4:5.3	54,500	9500	11,700	1.23
P7	40:40:20	N-MeAAm	THF	7	145	41	45.3:44.0:10.7	40,150	9600	11,700	1.22

^a^ determined by ^1^HNMR (P1, P6, P7) and HPLC (P2-P5). ^b^ theoretical molecular weight calculated using the following equation: M_n,th_ = ([M_1_]_0_ × M_W_M_1_ × X_1_ + [M_2_]_0_× M_W_M_2_ × X_2_)/[RAFT]_0_ + M_W_RAFT, where [M_1_]_0_, [M_2_]_0_, [RAFT]_0_, M_W_M_1_, M_W_M_2_, X_1_, X_2_, and M_W_RAFT correspond to initial concentrations of the monomers, RAFT agent, molar weights of the monomers, their conversions, and molar weight of RAFT agent. ^c^ Determined by GPC in THF with PSt standard calibration.

**Table 2 polymers-16-00134-t002:** Thermal characterization of copolymers.

Copolymer ID	Tg, °C	Tm, °C
P1 (MOEGM-AOEGM)	−68.8	−18.8
P2 (MOEGM-AOEGM-AAm)	−61.1	−16.8
P3 (MOEGM-AOEGM-AAm)	−62.2	−12.6/−3.8
P4 (MOEGM-AOEGM-MAAm)	−63.7	−20.2
P5 (MOEGM-AOEGM-MAAm)	−60.1	−19.4

**Table 3 polymers-16-00134-t003:** CMC and loading capacity of copolymers.

Copolymer ID	C_p_, °C	CMC (mg/mL)	Loading Capacity (mg Pyrene/g Polymer)
P1	56.1	1.35 × 10^−3^	22.1
P2	55.7	1.52 × 10^−3^	23.3
P3	39.7	1.55 × 10^−3^	23.4
P4	55.8	1.13 × 10^−3^	21.7
P5	59.0	1.26 × 10^−3^	21.0
P6	55.9	1.43 × 10^−3^	21.7
P7	53.7	1.26 × 10^−3^	20.3

**Table 4 polymers-16-00134-t004:** Hydrodynamic and molecular weight characteristics of the copolymers.

**ID**	Hydrodynamic Radius, R_h_, nm ^a^		$M_{W,ACN}$ ^b^	$A_{2,ACN}\cdot 10^{4},$ mole·cm^3^·g^−2 b^	$M_{W,H_{2}O}$ ^b^	$A_{2,H_{2}O}\cdot 10^{4},$ mole·cm^3^·g^−2 b^	N_agg_
	Acetonitrile	Water					
P1	8.4	5.8	224,500	0.42	223,300	0.31	1.0
P3	6.3; 21.0	6.7; 25	168,600	0.78	406,400	0.19	2.4
P2	5.4; 19.6	6.8; 26	150,500	0.05	413,300	−0.13	2.7
P4	6.3	5.0	95,600	1.45	166,200	−0.49	1.7
P5	6.2	3.7	80,000	1.61	144,400	−0.38	1.8
P6	6.2	5.0	104,700	1.07	180,200	−0.03	1.7
P7	6.8	5.8	106,500	−0.50	172,600	−0.31	1.6

^a^ hydrodynamic radii were determined by DLS at 25 and 27 °C in water and acetonitrile, respectively. ^b^ absolute weight average molecular weights and second virial coefficients determined by SLS in acetonitrile and water.

## Data Availability

The data presented in this study are available upon request from the corresponding author.

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
