# Peer review of "Amide-Containing Bottlebrushes via Continuous-Flow Photoiniferter Reversible Addition–Fragmentation Chain Transfer Polymerization: Micellization Behavior"

_polymers, 2023, doi:10.3390/polym16010134_

Round 1

Reviewer 1 Report

Comments and Suggestions for Authors

This paper reports the synthesis of ternary amphiphilic bottlebrush copolymers using photo-mediated RAFT polymerizations in a continuous flow photoreactor. It explores how monomer composition influences the copolymers' properties, including melting temperature, critical micelle concentration, dye loading capacity, and micelle aggregation number. I think this study offers valuable insights into new polymer micelle designs and should be considered for publication after minor revisions:

  1. The polymerization approach described appears quite similar to methodologies documented in previous literature. It would be beneficial to revise the title to better represent the focus on micelle properties of the synthesized bottlebrush copolymers, as this is a central theme of the study, in my opinion.
  2. notable point in Table 2 is the substantial difference between the theoretical and experimental Mw across all polymerization conditions. The manuscript would benefit from a brief explanation or discussion of these discrepancies.
  3. The polymerization rates of amide monomers relative to acrylate monomers raise a question: if the rates are not comparable, it could lead to a gradient structure within the synthesized polymer. This question needs a more comprehensive discussion in the manuscript, as it could significantly impact the structural and functional properties of the final polymer product.

Reviewer 2 Report

Comments and Suggestions for Authors

   In this manuscript, authors developed a novel method to synthesize MOEGM-AOEGM-amide ternary copolymers. It is an interesting work, and an effective method to carry out RAFT copolymers. After a minor revision, this manuscript could be accepted.

1.      1H NMR spectra of P1, P3, P5, and P7 needed to be provided to confirm their obtaining;

2.      The structures of these MOEGM-AOEGM-amide ternary copolymers could be provided;

3.      For CMC study, the MOEGM-AOEGM-amide ternary copolymers were used to fabricate nanoparticles (or nano-micelle). The size distribution and morphology of these nanoparticles could be provided;

4.      The full name of RAFT polymerization should be provided when it was first appeared;

Comments on the Quality of English Language

Minor editing of English language required
